# NRG-HN003: Phase I and Expansion Cohort Study of Adjuvant Pembrolizumab, Cisplatin and Radiation Therapy in Pathologically High-Risk Head and Neck Cancer

**DOI:** 10.3390/cancers13122882

**Published:** 2021-06-09

**Authors:** Julie E. Bauman, Jonathan Harris, Ravindra Uppaluri, Min Yao, Robert L. Ferris, Josephine Chen, Richard C. Jordan, Nikhil P. Joshi, Srinivas Jujjuvaparu, Dukagjin M. Blakaj, Christina Henson, Jawad Sheqwara, Loren K. Mell, Neilayan Sen, David A. Clump, Madhur K. Garg, Emrullah Yilmaz, Pedro Torres-Saavedra, Quynh-Thu Le

**Affiliations:** 1University of Arizona Cancer Center, University of Arizona, Tucson, AZ 85724, USA; 2NRG Oncology Statistics and Data Management Center, American College of Radiology, Philadelphia, PA 19102, USA; HarrisJ@NRGOncology.org (J.H.); TorresP@NRGOncology.org (P.T.-S.); 3Dana Farber Cancer Institute and Brigham and Women’s Hospital, Boston, MA 02115, USA; ruppaluri@partners.org; 4University Hospitals Seidman Cancer Center, Case Comprehensive Cancer Center, Case Western Reserve University, Cleveland, OH 44106, USA; min.yao@uhhospitals.org; 5UPMC Hillman Cancer Center, University of Pittsburgh, Pittsburgh, PA 15232, USA; ferrisrl@upmc.edu (R.L.F.); clumpda2@upmc.edu (D.A.C.); 6Department of Radiation Oncology, Kaiser Permanente, Dublin, CA 94568, USA; josephine.chen@kp.org; 7NRG Oncology Biospecimen Bank, University of California San Francisco, San Francisco, CA 94143, USA; richard.jordan@ucsf.edu; 8Cleveland Clinic Taussig Cancer Center, Cleveland, OH 44195, USA; Nikhil_Joshi@rush.edu; 9Illinois CancerCare PC, Peoria, IL 61615, USA; sjujjavarapu@illinoiscancercare.com; 10Ohio State University Comprehensive Cancer Center, The Ohio State University, Columbus, OH 43210, USA; dukagjin.Blakaj@osumc.edu; 11University of Oklahoma Health Sciences Center, University of Oklahoma, Oklahoma City, OK 73104, USA; Christina-Henson@ouhsc.edu; 12Henry Ford Health Institute, Detroit, MI 48202, USA; jsheqwa1@hfhs.org; 13University of California San Diego Moores Cancer Center, University of California, La Jolla, CA 92093, USA; lmell@ucsd.edu; 14Rush University Medical Center, Rush University, Chicago, IL 60612, USA; neilayan_sen@rush.edu; 15Montefiore Medical Center, Bronx, NY 10467, USA; mgarg@montefiore.org; 16University of New Mexico Cancer Center, University of New Mexico, Albuquerque, NM 87102, USA; YILMAZE@ccf.org; 17Department of Radiation Oncology, Stanford University, Stanford, CA 94305, USA; qle@stanford.edu

**Keywords:** head and neck cancer, pathologically high-risk, pembrolizumab, adjuvant, radiation therapy, cisplatin, phase I

## Abstract

**Simple Summary:**

The anti-PD1 monoclonal antibody pembrolizumab improves survival in recurrent/metastatic head and neck squamous cell carcinoma. Patients with locoregional, pathologically high-risk disease recur frequently despite adjuvant cisplatin–radiation therapy. Targeting PD1 may reverse immunosuppression induced by cancer, chemotherapy, or radiation therapy. We conducted a phase I trial with an expansion cohort (n = 20) to determine the recommended phase II schedule for adding fixed-dose pembrolizumab to standard adjuvant cisplatin–radiation therapy. Eligible patients had resected, human papillomavirus-negative, stage III–IV oral cavity, pharynx, or larynx cancer with extracapsular nodal extension or positive margin. A total of four dose-limiting toxicities were observed in 34 patients (fever, wound infection, diverticulitis, nausea). Three of four were successfully rechallenged with pembrolizumab. The recommended phase II schedule was declared as pembrolizumab 200 mg every 3 weeks for eight doses, starting one week before adjuvant CRT. The regimen was safe and feasible in the cooperative group setting. Further development is warranted.

**Abstract:**

The anti-PD1 monoclonal antibody pembrolizumab improves survival in recurrent/metastatic head and neck squamous cell carcinoma (HNSCC). Patients with locoregional, pathologically high-risk HNSCC recur frequently despite adjuvant cisplatin–radiation therapy (CRT). Targeting PD1 may reverse immunosuppression induced by HNSCC and CRT. We conducted a phase I trial with an expansion cohort (n = 20) to determine the recommended phase II schedule (RP2S) for adding fixed-dose pembrolizumab to standard adjuvant CRT. Eligible patients had resected HPV-negative, stage III–IV oral cavity, pharynx, or larynx HNSCC with extracapsular nodal extension or positive margin. RP2S was declared if three or fewer dose-limiting toxicities (DLT) occurred in a cohort of 12. DLT was defined as grade 3 or higher non-hematologic adverse event (AE) related to pembrolizumab, immune-related AE requiring over 2 weeks of systemic steroids, or unacceptable RT delay. A total of 34 patients enrolled at 23 NRG institutions. During the first cohort, only one DLT was observed (fever), thus RP2S was declared as pembrolizumab 200 mg every 3 weeks for eight doses, starting one week before CRT. During expansion, three additional DLTs were observed (wound infection, diverticulitis, nausea). Of the 34 patients, 28 (82%) received five or more doses of pembrolizumab. This regimen was safe and feasible in a cooperative group setting. Further development is warranted.

## 1. Introduction

Head and neck squamous cell carcinoma (HNSCC) is the 6th leading cancer worldwide. Among the nearly 800,000 incident cases of HNSCC estimated annually, approximately 80% are caused by environmental carcinogenesis including direct exposures to tobacco, alcohol, and/or areca nut [1]. Unlike human papillomavirus (HPV)-associated HNSCC, outcomes for HPV-negative HNSCC remain poor despite historical intensification approaches including altered radiation therapy (RT) fractionation [2], induction or adjuvant chemotherapy [3], or EGFR-targeted monoclonal antibodies (mAb) [4,5,6]. The current standards of care for adjuvant management of HPV-negative HNSCC are determined by pathologic risk. The adjuvant standard for patients who demonstrate a high-risk pathologic feature, specifically a positive surgical margin or extracapsular nodal extension (ENE), is concurrent cisplatin and RT (CRT), which improved disease-free survival (DFS) and locoregional control (LRC) compared with RT alone in the landmark European Organization for the Research and Treatment of Cancer (EORTC) 22931 and Radiation Therapy Oncology Group (RTOG) 9501 trials [7,8]. While EORTC 22931 but not RTOG 9501 was also positive for overall survival (OS) in the intent-to-treat population, unplanned subgroup analyses suggested that an OS benefit in both trials may be driven by patients with positive margin or ENE, and not with a sole risk factor of multiple positive nodes, the latter now classified as a pathologic intermediate-risk factor [9,10]. Despite the advance of CRT, patients with pathologically high-risk, HPV-negative disease have a 3-year DFS of only 30–50% [7,8,11], thus new intensification approaches represent a major unmet need.

Immunotherapy has become the “4th modality” in HNSCC, long recognized as an immunosuppressive disease [12]. Patients with HNSCC display tumor-permissive cytokine profiles, defective antigen presenting cells (APC), quantitative and qualitative T lymphocyte deficiencies, and frequent T cell expression of the co-inhibitory receptors CTLA-4 and programmed death-1 (PD1), so-called “immune checkpoints” [13,14,15,16,17,18,19,20,21,22,23]. Nivolumab and pembrolizumab, mAb against PD1, were shown to improve OS versus single-agent chemotherapy in platinum-refractory, recurrent/metastatic HNSCC, leading to their U.S. Food and Drug Administration approval in 2016 [24,25]. Long term follow-up confirmed better long-term survival irrespective of HPV status or PD-L1 expression [26]. In addition, pembrolizumab monotherapy or pembrolizumab plus platinum doublet chemotherapy improved OS relative to cetuximab plus platinum doublet, resulting in two first-line indications [27]. These practice-changing results raise the hypothesis that anti-PD1 mAb could improve outcomes in the high-risk, curative-intent setting if added to adjuvant CRT.

Ionizing RT induces adaptive immune responses via three broad mechanisms, which could be synergistic with immunotherapy: (1) immunogenic cell death, releasing tumor antigens for processing and presentation; (2) chemokine upregulation within the tumor microenvironment, recruiting T lymphocytes; (3) modulation of tumor phenotype, increasing expression of tumor antigens and major histocompatibility complex [28]. However, RT also induces local immune suppression by upregulation of PD-L1 on both tumor and myeloid-derived suppressor cells (MDSC), reducing the adaptive response and theoretically facilitating future relapse. In two syngeneic models, concurrent PD-L1 blockade and RT were synergistic in controlling tumor growth, and generated prolonged protective T cell immunity as demonstrated by subsequent abscopal effect [29]. The pro-immunogenic effects of radiation on cancer cells (e.g., the upregulation of MHC-1 and *Fas*) can be induced, at least in vitro, by radiation doses varying from about 2 Gy to 30 Gy [30]. In an immune competent HNSCC model, fractionated RT (2 Gy × 10) induced PD-L1 upregulation on tumor and infiltrating inflammatory cells as monitored by immunoPET/CT imaging using Zr-89 labeled anti-mouse PD-L1 mAb [31]. Moreover, the scheduling of anti-PD-L1 mAb was important for therapeutic outcome when given in combination with low dose fractionated radiation (2 Gy/fx) in three solid tumor preclinical models; specifically, concomitant administration of anti PD-L1 mAb and fractionated RT improved survival whereas sequential or adjuvant PD-L1 mAb after fractionated RT did not [32]. While the precise applicability of these preclinical models to the post-operative setting, where tumor is no longer in situ, is unknown, they provide guidance that conventionally fractionated RT is sufficiently immunogenic to be paired with anti-PD1 mAb and that anti-PD1 mAb exposure both prior to and concurrent with RT may be biologically and clinically important.

NRG-HN003 was a phase I study aimed at defining the recommended phase II schedule (RP2S) for the combination of fixed-dose pembrolizumab and adjuvant CRT in pathologically high-risk HNSCC, where a loading dose of pembrolizumab was administered and the subsequent overlap of pembrolizumab with CRT was maximized.

## 2. Materials and Methods

### 2.1. Human Subjects Considerations

This study was approved by the Central Institutional Review Board of the National Cancer Institute and conducted by the NRG Oncology group of the National Clinical Trials Network (NCTN). All subjects provided written informed consent. Primary inclusion criteria included: pathologically documented diagnosis of HNSCC involving the oral cavity (excluding lips), oropharynx (HPV-negative), hypopharynx or larynx; have undergone gross total surgical resection; the presence of at least one high-risk pathologic feature, specifically extracapsular nodal extension or a positive surgical margin (tumor on ink); pathologic stage III–IVb based upon the American Joint Commission on Cancer, version 7; in the case of oropharynx cancer, HPV-negative status as determined by p16 immunohistochemistry confirmed by central review; age ≥18; Zubrod performance status 0–1; adequate hematologic function, including absolute neutrophil count ≥1500/mm^3^ and platelets ≥100,000/mm^3^; creatinine clearance (CrCl) ≥50 mL/min; adequate hepatic function including serum total bilirubin ≤1.5 times the upper limit of normal (ULN), aspartate aminotransferase and alanine transaminase ≤ 2.5 × ULN; coagulation studies ≤ 1.5 × ULN; negative pregnancy test for women of child-bearing potential; no active autoimmune disease requiring a disease-modifying agent within the prior 2 years. Key exclusion criteria included: prior invasive cancer within 3 years; simultaneous primary or bilateral tumors; prior systemic therapy for the index cancer; prior radiation therapy that would result in an overlap with radiation fields for the study cancer; severe active co-morbidity; grade 3–4 electrolyte abnormalities.

### 2.2. Study Design and Statistical Considerations

This was an open-label, schedule-finding phase I trial with a planned expansion cohort, evaluating the safety and feasibility of the addition of a fixed dose of the anti-PD1 mAb pembrolizumab to standard adjuvant therapy with conventionally fractionated intensity-modulated RT (IMRT) and concurrent weekly cisplatin in patients with high-risk, HPV-negative HNSCC. The trial was designed to proceed in two stages: (1) the schedule-finding stage, during which patients would enroll to descending schedule levels, as shown in Table 1, and the recommended phase II schedule (RP2S) for the combination would be determined by the rate of dose-limiting toxicity (DLT); and (2) the expansion stage, where an expansion cohort of 20 patients would be treated at the RP2S in order to improve estimates of safety and feasibility within the NCTN setting.

At the starting level (schedule level 3), patients were treated with full dose pembrolizumab for 8 doses, starting with a loading dose the week prior to CRT. The DLT observation period was 11 weeks, starting with the first dose of pembrolizumab and extending 4 weeks post-IMRT. Eligible treated patients that had a DLT or completed the DLT observation period without a DLT were considered DLT-evaluable. A schedule level was considered unsafe should ≥4 DLTs occur. Should schedule level 3 enroll 12 DLT-evaluable patients with ≤3 DLTs, then that schedule would be selected for expansion and the de-escalation schedule levels would not be initiated. Depending upon the observed DLT rates, a minimum of 12 and a maximum of 36 patients would be enrolled to the schedule-finding stage. The subsequent expansion cohort of 20 patients would result in a total sample size ranging from 32 to 56. In all DLT-evaluable patients, a DLT rate <33% was considered acceptable (e.g., ≤10/32).

The primary endpoint of DLT was defined as the occurrence of a severe adverse event (AE) listed below, judged at least possibly related to pembrolizumab, and occurring during the specified observation period. DLTs were graded according to the Common Terminology Criteria for Adverse Events (CTCAE), version 4, and included: (1) any non-hematologic toxicity of grade 3 or higher except the following: grade 3 or 4 in-field radiation dermatitis for which IMRT was held ≤1 week (5 fractions); grade 3 or 4 mucositis for which IMRT was held ≤1 week; grade 3 or 4 hypomagnesemia, hypokalemia, or hypophosphatemia without life-threatening consequences correcting to grade 2 or lower with observation or replacement therapy; or grade 3 pain, dysphagia, weight loss, or fatigue; (2) grade 2 AST or ALT elevation >3 × ULN with concurrent elevation of total bilirubin >2 × ULN; (3) grade 3 or higher immune related adverse event (irAE) requiring systemic corticosteroids or other immunosuppressant for more than 2 weeks; (4) delay in completion of IMRT more than 2 weeks (10 fractions), or inability to complete prescribed IMRT course, due to immune toxicity at least possibly attributed to pembrolizumab; (5) neutropenia of grade 3 or higher with fever (oral temperature > 39 °C); (6) thrombocytopenia of grade 3 or higher with bleeding.

As defined, the probability of DLT for the standard combination of adjuvant CRT in patients with high-risk HNSCC is 5–10% [7,8], and for pembrolizumab monotherapy in patients with recurrent/metastatic HNSCC, it is 10% [25]. As such, the maximum acceptable DLT rate for this phase I combination was defined as the additive rate of 20%, which was considered clinically acceptable given the expected 50–70% rate of 3-year relapse in this high-risk population. For each schedule-finding cohort of 12 patients, the probability of being judged too toxic, when the true DLT rate was ≥45%, was at least 87%. If the true DLT rate was 20% or lower, the probability that a schedule level would be judged safe was ≥79%. Exact confidence intervals for the DLT rates were calculated using the Clopper–Pearson method. Although no restrictions were placed on the number of participating NRG sites, due to the importance of DLT assessment for subject safety and the primary endpoint, special requirements were in place. These included twice monthly toxicity data reporting, participation of local principal investigators in monthly DLT conference calls, and real time communication with the Study Chair regarding new AEs that may qualify as DLTs.

Secondary endpoints were analyzed in eligible treated patients (analyzable) and included one-year DFS, OS, local–regional failure (LRF), distant metastases (DM) rates, and rates of acute (≤180 days after the end of RT) and late (>180 days) toxicity. DFS and OS rates were estimated by the Kaplan–Meier method and LRF and DM rates by the cumulative incidence method.

### 2.3. Treatment Plan

The dose and frequency of the systemic therapies, cisplatin and pembrolizumab, are described per schedule level in Table 1. In all schedule levels, concurrent cisplatin was administered at 40 mg/m^2^/week. Weekly cisplatin at 40 mg/m^2^ is the current standard for adjuvant NRG CRT trials, as in the ongoing NRG/RTOG 1216, and was selected over bolus cisplatin as used in EORTC 22931 and RTOG 9501 due to the more linear, predictable toxicity profile facilitating phase I treatment intensification [33]. One dose reduction of cisplatin to 30 mg/m^2^ was permitted in the event of severe hematologic, renal, or neurologic toxicity; a subsequent event required discontinuation of cisplatin. No substitutions for cisplatin were allowed. No dose reductions of pembrolizumab were permitted. If a patient did not meet retreatment criteria for pembrolizumab due to a protocol-specified autoimmune toxicity, that dose was skipped. The definition of per-protocol cisplatin and pembrolizumab administration was receiving 85–115% of the planned dose, and acceptable variation was administration of <85% of the planned dose for protocol-specified reasons, in accordance with NRG standards for systemic therapy. IMRT was mandatory, and all participating institutions were credentialed by NRG Oncology for head and neck IMRT. Image-guided RT (IGRT) was optional, however mandatory if reduced margins were selected. IMRT was administered in 30 daily fractions of 2 Gy over 6 weeks, 5 fractions per week, using simultaneous integrated boost [60 Gy to the high-risk clinical target volume (CTV), 56 Gy to the lower risk CTV, and an optional 66 Gy to the highest risk CTV including the regions of positive margin or ENE].

## 3. Results

### 3.1. Patient and Tumor Characteristics

Twelve patients were enrolled to schedule level 3 from November 2016 through June 2017. The study re-opened to the expansion cohort from October 2017 and accrual was closed in October 2018 with a total of 37 patients enrolled and 34 analyzable patients, of whom 32 (12 phase I, 20 expansion cohort) were evaluable for DLT. Three enrolled patients were excluded from all analyses, two due to withdrawal of consent prior to starting any protocol treatment and one due to having gross residual disease following surgery (ineligible). Two analyzable patients were not DLT-evaluable due to: (1) refusing IMRT and cisplatin; (2) withdrawal of consent during the DLT observation period. Patients were enrolled at 23 NRG Oncology sites. Table 2 shows the distributions of patient and tumor characteristics for all analyzable patients. Median (min–max) age was 60 years (26–83); 67.6% of patients were male; 73.5% had Zubrod performance status of 1; 35.3% had >10 pack-years smoking history; 85.3% had an oral cavity primary site; 32.4% had pathologic T4 disease; 82.4% had pathologic N2–3 disease; 91.2% had extracapsular nodal extension (per central review); and 20.6% had a positive margin (per central review).

### 3.2. Toxicity

Table 3 summarizes acute grade 3–4 adverse events reported in 34 analyzable patients as (1) any relationship to treatment, (2) definitely, probably, or possibly related to treatment, and (3) definitely, probably, or possibly related to pembrolizumab; terms are sorted from highest to lowest incidence. Without regard to attribution, the highest grade reported was grade 4 in 13 patients (38.2%) and grade 3 in 19 patients (55.9%). The most common acute grade 3–4 adverse events were lymphocyte count decreased [18 (52.9%)], oral mucositis [16 (47.1%)], leukopenia [12 (35.3%)], dysphagia [12 (35.3%)], neutropenia [10 (29.4%)], and hyponatremia [10 (29.4%)]. Four events in four individual patients were considered possible irAE related to pembrolizumab: diverticulitis requiring antibiotics and up to 2 weeks of systemic steroids; wound infection complicated by persistent inflammation after antibiotic treatment; fever following the loading dose of pembrolizumab; and maculopapular rash requiring up to 2 weeks of systemic steroids. Except for the patient experiencing fever, all were successfully re-challenged with pembrolizumab. The patient experiencing colitis was evaluated by colonoscopy and found to have no evidence of autoimmune etiology, completed antibiotics, and was successfully re-challenged with pembrolizumab. One patient died of grade 5 sepsis, reported as unrelated to treatment, occurring 35 days after the end of radiation therapy and 19 days after the end of all treatment. Table 4 summarizes late grade 3–4 adverse events. Without regard to attribution, the highest grade reported was grade 4 in 2 patients (7.1%) and grade 3 in 6 patients (21.4%). The most common late grade 3–4 adverse events were weight loss [3 (10.7%)], dysphagia [2 (7.1%)], and anorexia [2 (7.1%)].

### 3.3. Protocol Treatment

Radiation Therapy: Of the 34 analyzable patients, 31 (91.2%) completed IMRT of 60–66 Gy in 30 fractions and 1 discontinued RT after 38 Gy (19 fractions) due to disease progression. Two patients refused CRT after the loading dose of pembrolizumab, withdrew consent, and were treated per standard of care off protocol. In the 32 patients who received RT, 23 (71.9%) received IGRT with reduced margins, 7 (21.9%) received IGRT with no reduced margins, and 2 (6.3%) did not receive IGRT. Central review indicated that RT was delivered per protocol or with acceptable variation in 29 (85.3%) patients (Appendix A). Of the 31 patients who completed RT, 8 received 60 Gy and 23 received 66 Gy to the highest risk CTV.

Cisplatin: Of the 34 analyzable patients, 32 received at least one dose of cisplatin, with 17 (50.0%) receiving all 6 doses, 10 (29.4%) receiving 5 doses, 4 (11.8%) receiving 4 doses, and 1 (2.9%) receiving 3 doses. Two patients refused cisplatin. Mean cisplatin dose was 197.6 mg/m^2^ out of a planned 240 mg/m^2^. Central review indicated cisplatin was delivered per protocol or with acceptable variation in 32 (94.1%) patients (Appendix A).

Pembrolizumab: All analyzable patients started pembrolizumab. Of the 34 patients, 28 (82.4%) received ≥ 5 doses of pembrolizumab, with the breakdown in number of doses received as follows: 8 doses (n = 17); 7 doses (n = 5); 6 doses (n = 4); 5 doses (n = 2); 4 doses (n = 1); 3 doses (n = 2); 2 doses (n = 1); 1 dose (n = 2). Mean pembrolizumab dose was 1288.2 mg out of a planned 1600 mg. Central review indicated that both concurrent and maintenance pembrolizumab were delivered per protocol or with acceptable variation in 32 (94.1%) patients (Appendix A).

### 3.4. Study Endpoints

Dose Limiting Toxicity: Among the 12 patients enrolled to schedule level 3 during the schedule-finding stage of the trial, 1 DLT (8.3%) was reported [95% confidence interval (CI): 0.2–38.5%]: grade 3 fever requiring hospitalization. This patient went off study and completed adjuvant CRT per standard of care. Per protocol, this DLT rate was acceptable and the expansion cohort was opened. During the expansion cohort, DLTs were reported for 3 additional patients: grade 3 anorexia and grade 3 wound infection requiring hospitalization; grade 3 diverticulitis requiring hospitalization and systemic steroids for two weeks or less; and grade 3 nausea. All three patients were successfully re-challenged with pembrolizumab. The overall DLT rate was 4 DLTs in 32 evaluable patients [12.5% (95% CI: 3.5–29.0%)], which met pre-specified criteria for safety and feasibility.

Oncologic Outcomes: Median follow-up for surviving patients was 1.5 years (min–max: 0.04–2.6). Three patients were censored for DFS with less than a year of follow-up, two of whom withdrew consent. There have been 13 DFS failures (4 local relapses, 1 regional, 2 local and regional, 5 distant, and 1 death without relapse) and 8 deaths reported. Among the 12 subjects who relapsed, 12 had ECE, 1 had a positive margin, and 6 were current or former smokers. The estimated 1-year DFS and OS rates were 62.3% (95% CI: 45.4–79.2%) and 81.1% (95% CI: 67.5–94.7%), respectively, as depicted in Figure 1A. The estimated 1-year LRF and DM rates are 18.7% (95% CI: 7.4–33.9%) and 15.9% (95% CI: 5.6–30.9%), respectively, as depicted in Figure 1B.

## 4. Discussion

The recommended phase II schedule for the combination of fixed-dose pembrolizumab with adjuvant CRT in patients with pathologically high-risk HNSCC is pembrolizumab 200 mg every 3 weeks for a total of eight doses starting the week prior to RT, IMRT 60–66 Gy delivered at 2 Gy/fraction five days per week, and cisplatin 40 mg/m^2^ weekly during IMRT. Based upon pre-specified criteria, this adjuvant regimen is safe and feasible in an NCTN protocol organization setting. The ability to maximally overlap pembrolizumab with full dose, adjuvant CRT, including a loading dose, two concurrent doses, and five maintenance doses, is generally in line with phase I studies in the definitive CRT setting. The safety of pembrolizumab, cisplatin 40 mg/m^2^/week, and definitive IMRT was demonstrated in 27 patients with locally advanced HNSCC in a phase IB trial [34]. Similarly, RTOG Foundation 3504 reported the safety of concurrent nivolumab, high dose bolus cisplatin (100 mg/m^2^ every 3 weeks), and definitive IMRT [35].

The necessity and duration of maintenance treatment with an anti-PD1 mAb in the adjuvant or definitive treatment setting is controversial in HNSCC. During trial design, no data were available to guide this decision. The developmental commitment to at least 1 year of anti-PD1/L1 mAb initially evolved in the recurrent/metastatic solid tumor setting, where gross disease is present and multiple cycles of immunomodulation may be required to penetrate the tumor and reprogram its immune microenvironment. In the context of definitive chemoradiation—either in HNSCC or in non-small cell lung cancer—1 year of maintenance immunotherapy also may fulfill this principle. However, the scientific rationale for 1 year of immunotherapy exposure in the adjuvant, minimal residual disease state is unclear. Prior institutional and NCTN protocol organization experience with the feasibility of 1 year of maintenance treatment with any drug had been poor. ECOG 3303 was a single-arm phase II study evaluating the combination of cisplatin, radiation, and cetuximab in locally advanced, unresectable HNSCC, where 1 year of maintenance cetuximab was incorporated for non-progressing patients [36]. Only 44 of 59 patients (75%) received any maintenance therapy, and median maintenance exposure was 5.5 months. In a phase III randomized trial of adjuvant cisplatin–radiation plus lapatinib vs. placebo in high-risk HNSCC, patients received lapatinib vs. placebo during chemoradiation and then for a 1-year maintenance period [37]. Only 60% of subjects received more than 9 months of maintenance lapatinib, a self-administered oral drug. In RTOG Foundation 3504, conducted contemporaneously with NRG-HN003, the administration of maintenance nivolumab was determined infeasible as only three of eight patients received seven of nine planned monthly doses, while the threshold for feasibility was four or more of eight [35]. In NRG-HN003, the decision to include eight total doses of pembrolizumab, including five maintenance doses for a total regimen length of 6 months, was pragmatic. Moreover, feasibility was described as composite for the whole regimen with no specific hypothesis test for the maintenance component. As 28 of 34 patients (82%) received five or more of eight planned doses of pembrolizumab, with delivery per protocol or acceptable variation in 94%, the regimen is considered feasible. However, the study does not provide evidence on whether the optimal number of doses of pembrolizumab was delivered; while the regimen is considered feasible, only half of patients received all eight planned doses. Of note, there was no evidence that the administration of pembrolizumab compromised the delivery of the curative-intent backbone regimen of cisplatin and RT, as the delivery of both standard components was in line with historical controls [7,8,11].

This is the first phase I multimodality study conducted by the NRG Oncology Head and Neck Committee and had no a priori limitation on number of accruing sites. Despite additional requirements for participation, including monthly conference calls to review DLTs, participants were enrolled at 23 distinct sites across the United States, including 12 sites accruing just one participant. Overall, the toxicity profile was manageable and no DLT precluded the completion of adjuvant CRT in this curative-intent population. This record of enrollment suggests that the conclusions of regimen safety and feasibility are generalizable to the community oncology setting.

NRG-HN003 was not designed to establish an efficacy signal in patients with pathologically high-risk HNSCC. The estimated 1-year DFS and OS rates are preliminary and descriptive. The broad 95% confidence intervals overlap with observed DFS and OS rates in RTOG 9501, EORTC 22931, and RTOG 0234, a randomized phase II trial evaluating adjuvant radiation therapy with docetaxel–cetuximab or docetaxel–cisplatin in high-risk HNSCC. However, unlike NRG-HN003, these historical trials were conducted prior to the current definitions of clinical and pathologic high risk. Lower risk cohorts were included, including those with HPV-positive disease or pathologic risk factors now considered intermediate-risk (two or more positive nodes without extracapsular extension; pT3 or T4 tumor status; perineural or lymphovascular invasion; level IV or V nodal involvement) [7,8,11]. As such, comparison of the preliminary efficacy of the NRG-HN003 regimen to historical trials would be invalid and misleading.

The use of the anti-PD1 mAb nivolumab and pembrolizumab has improved OS in patients with recurrent/metastatic HNSCC [24,25,26] and multiple other solid tumor indications, raising the hypothesis that treatment intensification with immune checkpoint inhibition could improve oncologic outcomes in high-risk, locally advanced HNSCC cohorts. Important limitations in the state of our knowledge regarding optimal timing and duration of anti-PD1/L1 mAb in the definitive and post-operative settings remain, particularly in light of the negative results from the phase III JAVELIN study evaluating the addition of concurrent and maintenance avelumab, an anti-PD-L1 mAb, to CRT in the definitive setting [38]. The perioperative phase III trial, MK-3475-689, is evaluating two neoadjuvant cycles of fixed-dose pembrolizumab followed by adjuvant pembrolizumab with RT or CRT depending upon pathologic risk, as compared to standard surgery followed by risk-stratified adjuvant RT or CRT. The primary endpoints are major pathological response to neoadjuvant pembrolizumab, as was observed in the phase II study [39], and event free survival. Mechanistically, this alternative approach may leverage immune recruitment and T cell priming while tumor is in situ and prior to the immunosuppressive effects of surgery, chemotherapy, and RT are observed. The effectiveness of PD1/L1 blockade during concurrent CRT may be less than suggested by preclinical models, where limited tumor volumes and no elective nodal radiation were employed, whereas greater target volumes and elective nodal radiation are required in the curative-intent management of HNSCC. In NRG-HN003, we have shown that the addition of a full dose anti-PD1 mAb to adjuvant CRT is safe and feasible in pathologically high-risk HPV-negative HNSCC, including a loading dose and maximal concomitant overlap. The efficacy of this approach requires randomized evaluation against the current standard of care. A similar regimen with the anti-PD-L1 mAb atezolizumab is now undergoing phase III testing against standard adjuvant CRT in NRG-RTOG 1216 (NCT01810913).

## 5. Conclusions

Patients with locoregionally-advanced HNSCC may be immunosuppressed from the cancer itself or its multimodality treatment with surgery, radiation therapy, and cisplatin chemotherapy. The anti-PD1 monoclonal antibody, pembrolizumab, improves survival in recurrent/metastatic head and neck squamous cell carcinoma. We hypothesize that PD1 checkpoint blockade may reverse immunosuppression caused by HNSCC or its treatment. This phase I study with expansion cohort was conducted to determine the RP2S of fixed-dose pembrolizumab when added to standard, adjuvant CRT. We observed only four DLTs, and immune-related AEs were rare. The recommended phase II schedule was declared as pembrolizumab 200 mg every 3 weeks for eight doses, starting 1 week before adjuvant CRT. The regimen was safe and feasible in the cooperative group setting, with 23 distinct sites accruing at least one subject. Further development of this regimen is warranted.

## Figures and Tables

**Figure 1 cancers-13-02882-f001:**
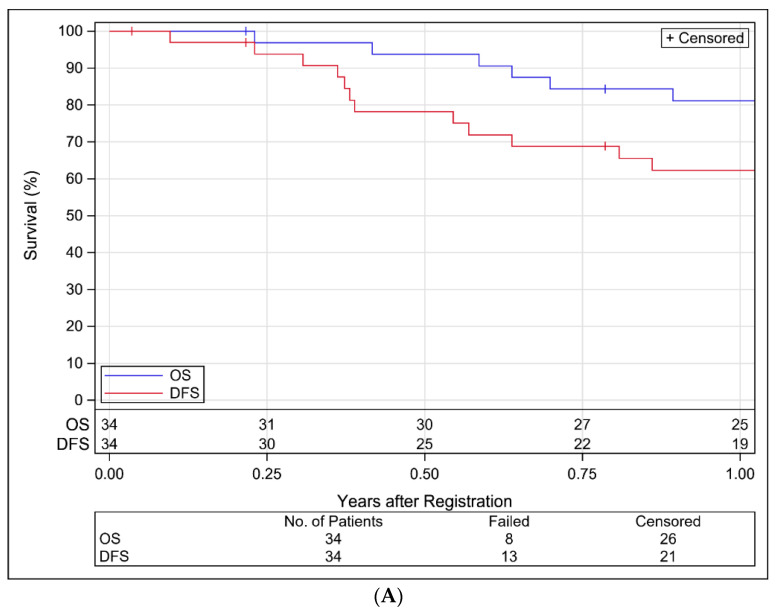
(**A**). Kaplan–Meier Estimates of Disease-Free and Overall Survival for Analyzable Patients in NRG-HN003. (**B**). Cumulative Incidence Estimates of Local-Regional Failure and Distant Metastasis for Analyzable Patients in NRG-HN003.

**Table 1 cancers-13-02882-t001:** Schedule Levels.

Modality	Week of Adjuvant IMRT													
	Loading	CRT						Maintenance						
	−1	1	2	3	4	5	6	9	12	15	18	21	24	27
IMRT (60 Gy, 2 Gy/Fx)—All		X	X	X	X	X	X							
Cisplatin 40 mg/m^2^ IV—All		X	X	X	X	X	X							
Pembrolizumab 200 mg IV														
Schedule 3 (Starting)	X			X			X	X	X	X	X	X		
Schedule 2 (1st de-escalation)	X						X	X	X	X	X	X	X	
Schedule 1 (2nd de-escalation)	X							X	X	X	X	X	X	X

**Table 2 cancers-13-02882-t002:** Patient and Tumor Characteristics for Analyzable Patients in NRG-HN003 (n = 34).

Characteristic	n	%
Age (years)		
≤65	27	79.4
>65	7	20.6
Gender		
Male	23	67.6
Female	11	32.4
Race		
American Indian/Alaska Native	1	2.9
Asian	1	2.9
Black or African American	2	5.9
White	29	85.3
Unknown or not reported	1	2.9
Ethnicity		
Hispanic or Latino	1	2.9
Not Hispanic or Latino	32	94.1
Unknown or not reported	1	2.9
Zubrod performance status		
0	9	26.5
1	25	73.5
Smoking history		
Never	14	41.2
Former	17	50.0
Current	3	8.8
Smoking history: pack-years		
≤10	22	64.7
>10	12	35.3
Primary site		
Oral cavity	29	85.3
Oropharynx, p16-negative	1	2.9
Hypopharynx	2	5.9
Larynx	2	5.9
Pathologic T stage		
pT1	5	14.7
pT2	10	29.4
pT3	8	23.5
pT4	11	32.4
Pathologic N stage		
pN0	1	2.9
pN1	5	14.7
pN2b	14	41.2
pN2c	8	23.5
pN3	6	17.6
Extracapsular nodal extension (per institution)		
No	4	11.8
Yes	30	88.2
Extracapsular nodal extension (central review)		
No	3	8.8
Yes	31	91.2
Positive margin (per institution)		
No	27	79.4
Yes	7	20.6
Positive margin (central review)		
No	27	79.4
Yes	7	20.6

**Table 3 cancers-13-02882-t003:** Summary of Grade 3–4 Acute Adverse Events for Analyzable Patients in NRG-HN003 (n = 34).

	Any Relationship		Related to Treatment		Related to Pembrolizumab	
Term	n (%) by Grade		n (%) by Grade		n (%) by Grade	
	3	4	3	4	3	4
Overall highest grade	19	13	20	11	12	7
	(55.9)	(38.2)	(58.8)	(32.4)	(35.3)	(20.6)
Lymphocyte count decreased	10	8	11	6	6	4
	(29.4)	(23.5)	(32.4)	(17.6)	(17.6)	(11.8)
Mucositis oral	15	1	13	1	4	0
	(44.1)	(2.9)	(38.2)	(2.9)	(11.8)	(0.0)
White blood cell decreased	8	4	7	4	0	3
	(23.5)	(11.8)	(20.6)	(11.8)	(0.0)	(8.8)
Dysphagia	12	0	8	0	3	0
	(35.3)	(0.0)	(23.5)	(0.0)	(8.8)	(0.0)
Neutrophil count decreased	4	6	3	4	1	1
	(11.8)	(17.6)	(8.8)	(11.8)	(2.9)	(2.9)
Hyponatremia	9	1	7	1	1	1
	(26.5)	(2.9)	(20.6)	(2.9)	(2.9)	(2.9)
Weight loss	7	0	5	0	2	0
	(20.6)	(0.0)	(14.7)	(0.0)	(5.9)	(0.0)
Anorexia	6	0	6	0	1	0
	(17.6)	(0.0)	(17.6)	(0.0)	(2.9)	(0.0)
Platelet count decreased	6	0	4	0	1	0
	(17.6)	(0.0)	(11.8)	(0.0)	(2.9)	(0.0)
Oral pain	4	0	3	0	0	0
	(11.8)	(0.0)	(8.8)	(0.0)	(0.0)	(0.0)
Dehydration	3	0	2	0	0	0
	(8.8)	(0.0)	(5.9)	(0.0)	(0.0)	(0.0)
Anemia	3	0	1	0	1	0
	(8.8)	(0.0)	(2.9)	(0.0)	(2.9)	(0.0)
Lung infection	1	1	1	1	0	0
	(2.9)	(2.9)	(2.9)	(2.9)	(0.0)	(0.0)
Fatigue	2	0	2	0	2	0
	(5.9)	(0.0)	(5.9)	(0.0)	(5.9)	(0.0)
Nausea	2	0	2	0	1	0
	(5.9)	(0.0)	(5.9)	(0.0)	(2.9)	(0.0)
Febrile neutropenia	2	0	2	0	0	0
	(5.9)	(0.0)	(5.9)	(0.0)	(0.0)	(0.0)
Infections and infestations—Other	2	0	1	0	1	0
	(5.9)	(0.0)	(2.9)	(0.0)	(2.9)	(0.0)
Dermatitis radiation	2	0	1	0	0	0
	(5.9)	(0.0)	(2.9)	(0.0)	(0.0)	(0.0)
Sore throat	2	0	1	0	0	0
	(5.9)	(0.0)	(2.9)	(0.0)	(0.0)	(0.0)
Soft tissue infection	2	0	0	0	0	0
	(5.9)	(0.0)	(0.0)	(0.0)	(0.0)	(0.0)
Pleural effusion	0	1	0	0	0	0
	(0.0)	(2.9)	(0.0)	(0.0)	(0.0)	(0.0)
Fever	1	0	1	0	1	0
	(2.9)	(0.0)	(2.9)	(0.0)	(2.9)	(0.0)
Lymphocyte count increased	1	0	1	0	1	0
	(2.9)	(0.0)	(2.9)	(0.0)	(2.9)	(0.0)
Pain	1	0	1	0	1	0
	(2.9)	(0.0)	(2.9)	(0.0)	(2.9)	(0.0)
Pharyngeal mucositis	1	0	1	0	1	0
	(2.9)	(0.0)	(2.9)	(0.0)	(2.9)	(0.0)
Rash maculo-papular	1	0	1	0	1	0
	(2.9)	(0.0)	(2.9)	(0.0)	(2.9)	(0.0)
Wound infection	1	0	1	0	1	0
	(2.9)	(0.0)	(2.9)	(0.0)	(2.9)	(0.0)
Dry mouth	1	0	1	0	0	0
	(2.9)	(0.0)	(2.9)	(0.0)	(0.0)	(0.0)
Gastrointestinal disorders—Other	1	0	1	0	0	0
	(2.9)	(0.0)	(2.9)	(0.0)	(0.0)	(0.0)
Metabolism and nutrition disorders—Other	1	0	1	0	0	0
	(2.9)	(0.0)	(2.9)	(0.0)	(0.0)	(0.0)
Pharyngolaryngeal pain	1	0	1	0	0	0
	(2.9)	(0.0)	(2.9)	(0.0)	(0.0)	(0.0)
Vomiting	1	0	1	0	0	0
	(2.9)	(0.0)	(2.9)	(0.0)	(0.0)	(0.0)
Aphonia	1	0	0	0	0	0
	(2.9)	(0.0)	(0.0)	(0.0)	(0.0)	(0.0)
Back pain	1	0	0	0	0	0
	(2.9)	(0.0)	(0.0)	(0.0)	(0.0)	(0.0)
Catheter-related infection	1	0	0	0	0	0
	(2.9)	(0.0)	(0.0)	(0.0)	(0.0)	(0.0)
Colitis	1	0	0	0	0	0
	(2.9)	(0.0)	(0.0)	(0.0)	(0.0)	(0.0)
Diarrhea	1	0	0	0	0	0
	(2.9)	(0.0)	(0.0)	(0.0)	(0.0)	(0.0)
Generalized muscle weakness	1	0	0	0	0	0
	(2.9)	(0.0)	(0.0)	(0.0)	(0.0)	(0.0)
Hepatobiliary disorders—Other	1	0	0	0	0	0
	(2.9)	(0.0)	(0.0)	(0.0)	(0.0)	(0.0)
Hypertension	1	0	0	0	0	0
	(2.9)	(0.0)	(0.0)	(0.0)	(0.0)	(0.0)
INR increased	1	0	0	0	0	0
	(2.9)	(0.0)	(0.0)	(0.0)	(0.0)	(0.0)
Mucosal infection	1	0	0	0	0	0
	(2.9)	(0.0)	(0.0)	(0.0)	(0.0)	(0.0)
Musculoskeletal and connective tissue disorder—Other	1	0	0	0	0	0
	(2.9)	(0.0)	(0.0)	(0.0)	(0.0)	(0.0)
Skin and subcutaneous tissue disorders—Other	1	0	0	0	0	0
	(2.9)	(0.0)	(0.0)	(0.0)	(0.0)	(0.0)
Trismus	1	0	0	0	0	0
	(2.9)	(0.0)	(0.0)	(0.0)	(0.0)	(0.0)
Upper respiratory infection	1	0	0	0	0	0
	(2.9)	(0.0)	(0.0)	(0.0)	(0.0)	(0.0)
Vasovagal reaction	1	0	0	0	0	0
	(2.9)	(0.0)	(0.0)	(0.0)	(0.0)	(0.0)
Weight gain	1	0	0	0	0	0
	(2.9)	(0.0)	(0.0)	(0.0)	(0.0)	(0.0)

Adverse events (AE) were graded with CTCAE, version 4. AEs are sorted from highest to lowest incidence. AEs are reported as worst grade per patient within each category and relationship term. Acute: ≤180 days from the end of radiation therapy. Related: definitely, probably, or possibly related. Includes AEs where relationship to treatment was missing.

**Table 4 cancers-13-02882-t004:** Summary of Grade 3–4 Late Adverse Events for Analyzable Patients in NRG-HN003 with >180 Days of Follow-Up Post-IMRT (n = 28).

	Any Relationship		Related to Treatment		Related to Pembrolizumab	
Term	n (%) by Grade		n (%) by Grade		n (%) by Grade	
	3	4	3	4	3	4
Overall highest grade	6	2	5	1	3	0
	(21.4)	(7.1)	(17.9)	(3.6)	(10.7)	(0.0)
Weight loss	3	0	2	0	0	0
	(10.7)	(0.0)	(7.1)	(0.0)	(0.0)	(0.0)
Dysphagia	2	0	1	0	1	0
	(7.1)	(0.0)	(3.6)	(0.0)	(3.6)	(0.0)
Anorexia	2	0	1	0	0	0
	(7.1)	(0.0)	(3.6)	(0.0)	(0.0)	(0.0)
Hyponatremia	0	1	0	1	0	0
	(0.0)	(3.6)	(0.0)	(3.6)	(0.0)	(0.0)
Hypercalcemia	0	1	0	0	0	0
	(0.0)	(3.6)	(0.0)	(0.0)	(0.0)	(0.0)
Hypophosphatemia	0	1	0	0	0	0
	(0.0)	(3.6)	(0.0)	(0.0)	(0.0)	(0.0)
Esophageal stenosis	1	0	1	0	1	0
	(3.6)	(0.0)	(3.6)	(0.0)	(3.6)	(0.0)
Pharyngeal mucositis	1	0	1	0	1	0
	(3.6)	(0.0)	(3.6)	(0.0)	(3.6)	(0.0)
Fracture	1	0	1	0	0	0
	(3.6)	(0.0)	(3.6)	(0.0)	(0.0)	(0.0)
Mucositis oral	1	0	1	0	0	0
	(3.6)	(0.0)	(3.6)	(0.0)	(0.0)	(0.0)
Infections and infestations—Other	1	0	0	0	1	0
	(3.6)	(0.0)	(0.0)	(0.0)	(3.6)	(0.0)
Abdominal pain	1	0	0	0	0	0
	(3.6)	(0.0)	(0.0)	(0.0)	(0.0)	(0.0)
Anemia	1	0	0	0	0	0
	(3.6)	(0.0)	(0.0)	(0.0)	(0.0)	(0.0)
Aspiration	1	0	0	0	0	0
	(3.6)	(0.0)	(0.0)	(0.0)	(0.0)	(0.0)
Dehydration	1	0	0	0	0	0
	(3.6)	(0.0)	(0.0)	(0.0)	(0.0)	(0.0)
Dermatitis radiation	1	0	0	0	0	0
	(3.6)	(0.0)	(0.0)	(0.0)	(0.0)	(0.0)
Fall	1	0	0	0	0	0
	(3.6)	(0.0)	(0.0)	(0.0)	(0.0)	(0.0)
Hypertension	1	0	0	0	0	0
	(3.6)	(0.0)	(0.0)	(0.0)	(0.0)	(0.0)
Ileus	1	0	0	0	0	0
	(3.6)	(0.0)	(0.0)	(0.0)	(0.0)	(0.0)
Vomiting	1	0	0	0	0	0
	(3.6)	(0.0)	(0.0)	(0.0)	(0.0)	(0.0)

AEs were graded with CTCAE, version 4. AEs are sorted from highest to lowest incidence. AEs are reported as worst grade per patient within each category and relationship term. Late: >180 days from the end of radiation therapy. Related: definitely, probably, or possibly related. Includes AEs where relationship to treatment was missing.

## Data Availability

All published data from this paper will be available upon request in accordance with NRG Oncology’s data sharing policy, which can be found at https://www.nrgoncology.org/Resources/Ancillary-Projects-Data-Sharing-Application (accessed on 7 June 2021).

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
