# Peer review of "NRG-HN003: Phase I and Expansion Cohort Study of Adjuvant Pembrolizumab, Cisplatin and Radiation Therapy in Pathologically High-Risk Head and Neck Cancer"

_cancers, 2021, doi:10.3390/cancers13122882_

Round 1

Reviewer 1 Report

The authors conducted a phase I trial to investige different dose regimens for pembeolizumab in the setting of adjuvant chemoradiotherapy for HNSCC. The paper is well written and of great interest.

Minor comments:

The authors mainly refer to a maximum dose of 60Gy but also state that 66Gy were optional. I would suggest to use the dose description of 60-66Gy throughout the manuscript as also recommended by NCCN guidelines and please provide numbers of how many patients received 60 or 66 Gy (I might have missed this).

You conclude that Pembro 200mg q3w for eight cycles is optimal. Please discuss ongoing trial such as MK-3475-689 as they use different schedules.

Reviewer 2 Report

Abstract

No comments

Introduction

-concise and well written

Methods

-DLTs appear sufficient and appropriate

-Authors mention 23 NRG sites participated in the study. With so many sites involved in a phase I study, how did the PI/NRG ensure DLTs were reported appropriately and in a timely manner?

Results

-Authors report RT compliance and dosing.  Table 2 shows 31/34 patients had ENE per central review and 7 had positive margins.  Typically these regions are given 66Gy based on sufficient data.  Can the authors explain why all patients received only 60Gy?  The protocol states that 66Gy was not mandatory but most NRG facilities will routinely give 66Gy.  Please explain

Reviewer 3 Report

The manuscript entitled NRG-HN003: Phase I and Expansion Cohort Study of Adjuvant Pembrolizumab, Cisplatin and Radiation Therapy in Pathologically High-Risk Head and Neck Cancer reports the results of a phase I trial with expansion cohort in 34 patients and concludes that the regimen of pembrolizumab 200 mg every 3 weeks for 8 doses, starting one week before adjuvant concurrent chemo-radiation therapy with weekly cisplatin 40 mg/m2 for pathologically high-risk head and neck cancer patients was safe and feasible. This study was well designed and the results are clearly presented.

There are two minor critiques:

1. In Page 4 Table 1, there are some mistakes in the lines of Schedule 3, 2 and 1.

2. In Page 7 Table 3, at the first and second row, the number of "Any relationship" should be higher than the number of " Related to treatment".
